# Beyond Independent Frames: Latent Attention Masked Autoencoders for Multi-View Echocardiography

**Simon Böhi**[1],[*] **Irene Cannistraci**[2], **Sergio Muñoz Gonzalez**[1], **Moritz Vandenhirtz**[2],
**Sonia Laguna**[2], **Samuel Ruiperez-Campillo**[2], **Max Krähenmann**[1], **Andrea Agostini**[2],
**Ece Ozkan**[1],[†] **Thomas M. Sutter**[2],[†] **Julia E. Vogt**[2],[†]
[1]Department of Biomedical Engineering, University of Basel, Switzerland
[2]Department of Computer Science, ETH Zurich, Switzerland

## Abstract

Echocardiography is a widely used modality for cardiac assessment due to its non-invasive and cost-effective nature, but the sparse and heterogeneous spatiotemporal views of the heart pose distinct challenges. Existing masked autoencoder (MAE) approaches typically process images or short clips independently, failing to capture the inherent multi-view structure required for coherent cardiac representation. We introduce *Latent Attention Masked Autoencoder* (LAMAE), a foundation model architecture tailored to the multi-view nature of medical imaging. LAMAE augments the standard MAE with a latent attention module that enables information exchange across frames and views directly in latent space. This allows the model to aggregate variable-length sequences and distinct views, reconstructing a holistic representation of cardiac function from partial observations. We pretrain LAMAE on MIMIC-IV-ECHO, a large-scale, uncurated dataset reflecting real-world clinical variability. To the best of our knowledge, we present the first results for predicting ICD-10 codes from MIMIC-IV-ECHO videos. Furthermore, we empirically demonstrate that representations learned from adult data transfer effectively to pediatric cohorts despite substantial anatomical differences. These results provide evidence that incorporating structural priors, such as multi-view attention, yields significantly more robust and transferable representations.

## 1 Introduction

Cardiovascular diseases remain the leading cause of mortality worldwide, making timely and accurate assessment of cardiac structure and function critical (Ozkan et al., 2024; Ouyang et al., 2020). Echocardiography plays a central role in this assessment due to its non-invasive nature, real-time imaging capability, and relatively low cost (Dohi, 2019; Stebler et al., 2025), resulting in one of the most widely used imaging modalities in clinical cardiology. However, echocardiograms are inherently noisy (Kang et al., 2023). Their interpretation requires substantial domain expertise, and measurements are subject to both inter- and intra-observer variability (Ozkan et al., 2024; Ouyang et al., 2020). These challenges make echocardiography a natural target for machine learning methods aimed at improving robustness, consistency, and scalability of cardiac assessment (Michel et al., 2025; Nazari et al., 2025; Mor-Avi et al., 2023).

Self-supervised representation learning has shown strong transfer performance across vision tasks by learning general-purpose features from large amounts of unlabeled data. In particular, Masked Autoencoders (MAEs) learn effective visual representations by reconstructing masked portions of the input and have become a competitive and scalable pretraining strategy for images (He et al., 2021). Video extensions such as VideoMAE (Tong et al., 2022; Wang et al., 2023) apply similar masking objectives to spatiotemporal "tubes", enabling representation learning directly from raw video data. MAE-based approaches have been successfully applied to ultrasound images

---

[*]Correspondence to `simon.boehi@unibas.ch`
[†]Shared senior authorship

(Jiao et al., 2024; Megahed et al., 2026; Kang et al., 2023; 2026) and to full echocardiography videos (Kim et al., 2025; Stebler et al., 2025; Zhang et al., 2024; Yang et al., 2026). Prior work extend MAE or VideoMAE with additional inductive biases, including temporal alignment losses (Kim et al., 2025; Yang et al., 2026; Stebler et al., 2025) and noise- or blur-based reconstruction strategies to improve latent representations (Kang et al., 2023; 2026; Yang et al., 2026). However, most of these approaches operate on a single image or a single video clip, which differs from clinical practice, where echocardiography studies typically consist of multiple videos acquired from different views, each capturing complementary anatomical information. Some recent works address this multi-view setting. Mokhtari et al. (2023) propose a hierarchical transformer to integrate multiple echocardiography videos, but train task-specific models from scratch without self-supervised pretraining. Tohyama et al. (2025) introduce a MAE-based method operating on the latent representations of multiple video encoders, but rely on frozen per-view embeddings from a pretrained EchoPrime encoder Vukadinovic et al. (2025), inherently limiting performance to the capabilities of the frozen encoder.

In this work, we introduce the *Latent Attention Masked Autoencoder (LAMAE)*, a self-supervised pretraining framework designed to flexibly handle ultrasound images, videos, and multi-view echocardiography studies. LAMAE introduces a Latent Attention (LA) module that enables information exchange across frames and views during pretraining, while retaining the simplicity of the standard MAE reconstruction objective. Our contributions are three-fold: (i) we propose a latent-attention MAE architecture designed to flexibly handle heterogeneous, multi-view echocardiography data; (ii) we provide the first evaluation of ICD-10 code prediction from echocardiography videos in MIMIC-IV-ECHO (Gow et al., 2023); and (iii) we demonstrate strong transfer of the learned representations to EchoNet-Dynamic (Ouyang et al., 2020) and EchoNet-Pediatrics (Reddy et al., 2023). Performance remains strong on pediatric echocardiography, where anatomical and disease differences challenge models trained primarily on adult data (Reddy et al., 2023).

Figure 1: **LAMAE architecture overview**. During pretraining (*left*), masked frames from multiple views are encoded and fused through the Latent Attention (LA) module to learn shared representations, which are used to reconstruct full frames. During finetuning (*right*), frames are processed through that same encoder and LA module, followed by a lightweight classification head.

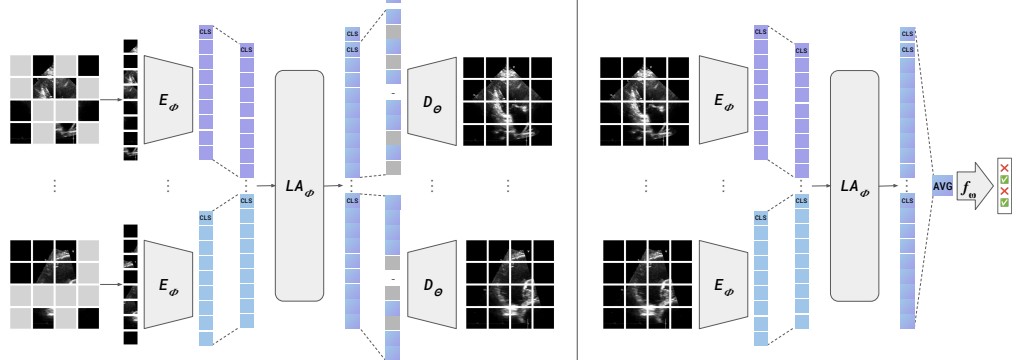

## 2 METHODS

We consider an echocardiography dataset $\mathbb{X} = \{\boldsymbol{X}^{(i)}\}_{i=1}^{N}$. Each study $\boldsymbol{X}^{(i)}$ consist of a set of views indexed by $\mathbb{V}^{(i)}$, and each view $j$ contains a sequence of frames $\mathbb{F}^{(i,j)}$. We denote an individual frame as $\boldsymbol{x}_{j,k}^{(i)}$, where $j$ indexes the view and $k$ indexes the frame.

We propose to extend the MAE framework (He et al., 2021) to the echocardiography domain. Our core contribution is the *Latent Attention* (LA) module, inserted between the encoder $E_\phi$ and the decoder $D_\theta$. As illustrated in Figure 1, this module enables the model to learn correlations and shared information across different views and frames via a self-attention mechanism, while maintaining the flexibility to process independent instances when necessary (Ilse et al., 2018; Lee et al., 2019).

Each frame $\boldsymbol{x}_{j,k}$ is processed independently by an encoder $E_\phi$. Following standard MAE practice, a random subset of image patches is masked, and only the set of visible tokens $\mathcal{T}_{\text{vis}}$ is fed to the

encoder. We denote the masking operation by $M(\cdot)$, with masking ratio $\alpha$. The encoder produces a sequence of latent patch tokens $\boldsymbol{z}_{j,k} = E_\phi(M_E(\boldsymbol{x}_{j,k}))$. We concatenate these tokens across all sampled views and frames into a single set:

$$\boldsymbol{Z}^{(i)} = \{\boldsymbol{z}_{j,k}^{(i)} | j \in \tilde{\mathbb{V}}^{(i)}, k \in \tilde{\mathbb{F}}^{(i,j)}\} , \tag{1}$$

where $\tilde{\mathbb{V}}^{(i)} \subseteq \mathbb{V}^{(i)}$ and $\tilde{\mathbb{F}}^{(i,j)} \subseteq \mathbb{F}^{(i,j)}$ are sampled subsets of views and frames.

Additional random masking is applied in latent space. The output of the LA module is

$$\boldsymbol{Z}_{out}^{(i)} = LA_\phi(M_{\text{LA}}(\boldsymbol{Z}^{(i)})) . \tag{2}$$

We reconstruct only masked patches and average the reconstruction error across frames, views, and masked tokens.

$$\mathcal{L}\left(\boldsymbol{X}^{(i)}\right) = \sum_{j \in \tilde{\mathbb{V}}^{(i)}} \sum_{k \in \tilde{\mathbb{F}}^{(i,j)}} \sum_{t \notin \mathcal{T}_{\text{vis}}} \frac{1}{\alpha_{\text{E}}} \frac{1}{|\tilde{\mathbb{V}}^{(i)}|} \frac{1}{|\tilde{\mathbb{F}}^{(i,j)}|} \left\| \boldsymbol{x}_{j,k_t}^{(i)} - \hat{\boldsymbol{x}}_{j,k_t}^{(i)} \right\|_2^2 , \text{ where } \hat{\boldsymbol{x}}_{j,k}^{(i)} = D_\theta(\boldsymbol{Z}_{\text{out}}^{(i)})_{j,k}$$

We consider two variants of the model. The frame-based variant, denoted **LAMAE**, processes frames independently using a standard image encoder. In the video-based variant, **Video-LAMAE**, we replace the frame-based encoder and decoder with their spatiotemporal counterparts that operate on clips. In both settings, the LA module operates on the resulting latent tokens in the same manner. Similar to VideoMAE, encoder and decoder operate at different embedding dimensionalities. We apply a linear projection after the LA module to align encoder-decoder dimensionalities.

For downstream finetuning, we average the latent tokens

$$\bar{\boldsymbol{Z}}_{out}^{(i)} = \sum_{j \in \tilde{\mathbb{V}}^{(i)}} \sum_{k \in \tilde{\mathbb{F}}^{(i,j)}} \frac{1}{|\tilde{\mathbb{V}}^{(i)}|} \frac{1}{|\tilde{\mathbb{F}}^{(i,j)}|} \boldsymbol{z}_{out}{}_{j,k}^{(i)} , \tag{3}$$

and pass it to a simple multilayer perceptron head for classification or regression.

## 3 EXPERIMENTS AND RESULTS

**Dataset and preprocessing.** We pretrain LAMAE on the MIMIC-IV-ECHO dataset. From the over 500'000 individual echocardiograms, we select only 2D B-mode ultrasound videos and exclude Doppler data and single-image files for simplicity. Extending LAMAE to additional modalities is straightforward and left for future work. Dataset details are summarized in Appendix Table 3. Video preprocessing follows the EchoPrime pipeline (Vukadinovic et al., 2025). We link MIMIC-IV-ECHO with MIMIC-IV (Johnson et al., 2024) to obtain ICD-10 discard codes, then the 40 most prevalent codes are selected as prediction targets. Details are provided in Appendix Section 5.1.

**Pretraining.** We pretrain four models with identical training settings: LAMAE, Video-LAMAE, and the standard baselines VideoMAE and MAE. All models are pretrained on MIMIC-IV-ECHO using a Vision Transformer (ViT)-Base encoder and a ViT-Tiny decoder; the LA module consists of 3 layers. For each study, we sample 8 views. Training details are provided in Appendix Section 5.2.

**Finetuning.** We finetune all pretrained models for 120 epochs on studies with available ICD-10 codes to perform multi-label classification of the 40 selected ICD-10 codes using two regimes: (i) full finetuning and (ii) frozen-backbone, where only the classification head is trained. Table 1 shows mean $\pm$ std AUROC and F1 scores over three seeds; see Appendix Section 5.3 for metric details.

Under full finetuning, all models achieve comparable performance indicating that the task is challenging but that all pretraining strategies learn useful representations. Nevertheless, LAMAE and Video-LAMAE consistently achieve higher AUROC and F1 scores, than their standard MAE counterparts (e.g., 0.75 vs 0.73 AUROC for video models), demonstrating the value of the LA module in aggregating multi-view information.

In the frozen backbone setting, while AUROC scores plateau around 0.62, we observe a distinct advantage for spatiotemporal models in terms of F1 score. Video-based architectures outperform

Table 1: **ICD-10 code prediction.** Average AUROC and F1 scores for full finetuning and frozen-backbone settings (mean ± std over three seeds). Best results are in bold, second best are underlined.

| Method | Full Finetuning | | Frozen Backbone | |
|---|---|---|---|---|
| | AUROC | F1 | AUROC | F1 |
| Image-MAE | 0.72 ± 0.02 | 0.58 ± 0.04 | **0.62** ± 0.02 | 0.18 ± 0.01 |
| VideoMAE | 0.73 ± 0.02 | 0.58 ± 0.03 | 0.61 ± 0.02 | **0.28** ± 0.03 |
| LAMAE (ours) | 0.74 ± 0.02 | 0.59 ± 0.04 | **0.62** ± 0.02 | 0.20 ± 0.01 |
| Video-LAMAE (ours) | **0.75** ± 0.01 | **0.60** ± 0.03 | **0.62** ± 0.02 | 0.27 ± 0.03 |

frame-based ones (i.e., 0.27–0.28 vs. 0.18–0.20), suggesting that temporal features are particularly robust for diagnosis when the feature extractor is fixed. Finally, the performance gap between finetuning and frozen settings highlights the necessity of end-to-end adaptation for this specific task. Figure 2 details the AUROC results for the top 10 ICD-10 codes, selected based on the highest mean F1 scores across all finetuning runs. These results reveal that performance gains over the baselines are not uniform across diagnoses. We hypothesize that the codes showing the largest improvements are those that rely most heavily on information aggregated across multiple views. Further investigation is required to confirm this. Additional results are reported in Appendix Section 5.4.

Figure 2: Per-code AUROC results for the 10 top-performing ICD-10 codes under full finetuning.

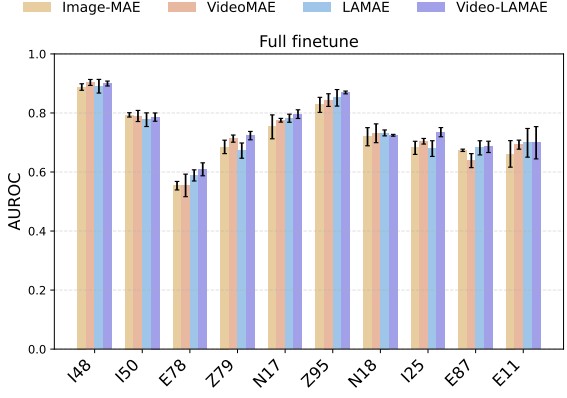

**Transfer.** We evaluate transfer performance on EchoNet-Dynamics and EchoNet-Pediatrics by finetuning pretrained models for 60 epochs with a regression head to predict Left Ventricular Ejection Fraction (LVEF). Table 2 reports the Mean Absolute Error (MAE) for both settings.

Table 2: **Transfer performance on LVEF prediction.** Results are reported as MAE under finetuning and frozen-backbone settings. Best results are in bold, second best are underlined.

| Method | EchoNet-Dynamics | | EchoNet-Pediatrics | |
|---|---|---|---|---|
| | Full Finetuning | Frozen Backbone | Full Finetuning | Frozen Backbone |
| Image-MAE | 5.27 ± 0.11 | 8.14 ± 0.05 | 4.92 ± 0.08 | 6.80 ± 0.04 |
| VideoMAE | **4.34** ± 0.07 | 6.92 ± 0.20 | 4.24 ± 0.13 | 6.61 ± 0.16 |
| LAMAE (ours) | 4.38 ± 0.09 | 7.40 ± 0.04 | 4.22 ± 0.06 | 6.73 ± 0.10 |
| Video-LAMAE (ours) | **4.34** ± 0.04 | **6.78** ± 0.13 | **3.92** ± 0.05 | **6.49** ± 0.09 |

EchoNet-Dynamics is a single-view dataset, restricting models to temporal integration only. As expected, VideoMAE and Video-LAMAE achieve nearly identical performance (4.34 MAE). However, LAMAE significantly outperforms Image-MAE (4.38 vs. 5.27) and matches the performance of the heavier spatiotemporal VideoMAE. This confirms that the LA module effectively captures temporal dynamics even when using a standard 2D image encoder.

EchoNet-Pediatrics presents a more challenging scenario: it contains multiple views and represents a domain shift to pediatric patients. Here, the benefits of our method are most pronounced. Here Video-LAMAE achieves the best overall performance (3.92 MAE), reducing the error by over 7% compared to VideoMAE. This substantial gap validates our core hypothesis: the LA module successfully aggregates information across different views, which standard video models cannot do. Additionally, Video-LAMAE shows the best robustness in the frozen setting, indicating that the multi-view representations learned during pretraining generalize well to out-of-distribution data.

## 4 CONCLUSION AND FUTURE WORK

We introduced LAMAE, a foundation model designed to handle multi-view and heterogeneous echocardiography data. We provided the first evaluation of ICD-10 code prediction on the MIMIC-IV-ECHO dataset and analyzed which clinical codes are predictable from imaging data alone. Compared to image- and video-based MAE baselines, LAMAE and Video-LAMAE show improved ICD-10 code prediction and superior transfer performance, particularly under domain shift and multi-view conditions. Future work will focus on improving pretraining strategies through explicit cross-frame or cross-view reconstruction, the incorporation of Doppler videos and single-frame studies, and the development of more scalable attention mechanisms to better exploit the complex structure of real-world echocardiography data.

## 5 ACKNOWLEDGEMENTS

This work was supported under project ID a135, a150 as part of the Swiss AI Initiative, through a grant from the ETH Domain and computational resources provided by the Swiss National Supercomputing Centre (CSCS) under the Alps infrastructure.

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

APPENDIX

## 5.1 DATASET AND PREPROCESSING

MIMIC-IV-ECHO does not provide native clinical labels, but $\approx 48\%$ of studies can be linked to hospital stays in MIMIC-IV (Johnson et al., 2024), each annotated with ICD-10 discard codes covering diagnoses, symptoms, and related clinical conditions. Since some ICD-10 codes (e.g., mental and behavioural disorders) are unlikely to be predictable from echocardiography alone, we adopt the following design choices. First, we normalize all ICD-10 codes to the third hierarchy level using the `simple_icd_10_cm` Python library (Travasci, 2025), balancing clinical specificity with sufficient code prevalence. Then, we select the 40 most prevalent ICD-10 codes (minimum prevalence $\approx 10\%$) as prediction targets. The selected codes and prevalence are listed in Section 5.1.

| ICD-10 Code | Description | Prevalence (%) |
|---|---|---|
| A41 | Other sepsis | 13.55 |
| D62 | Acute posthemorrhagic anemia | 12.54 |
| D63 | Anemia in chronic diseases classified elsewhere | 11.26 |
| D64 | Other anemias | 13.06 |
| D69 | Purpura and other hemorrhagic conditions | 14.02 |
| E03 | Other hypothyroidism | 17.83 |
| E11 | Type 2 diabetes mellitus | 37.99 |
| E66 | Overweight and obesity | 14.40 |
| E78 | Disorders of lipoprotein metabolism and other lipidemias | 56.11 |
| E87 | Other disorders of fluid, electrolyte and acid-base balance | 35.02 |
| F32 | Depressive episode | 19.43 |
| F41 | Other anxiety disorders | 16.61 |
| G47 | Sleep disorders | 20.83 |
| I10 | Essential (primary) hypertension | 28.10 |
| I11 | Hypertensive heart disease | 18.91 |
| I13 | Hypertensive heart and chronic kidney disease | 22.83 |
| I21 | Acute myocardial infarction | 16.11 |
| I25 | Chronic ischemic heart disease | 40.81 |
| I27 | Other pulmonary heart diseases | 15.53 |
| I48 | Atrial fibrillation and flutter | 39.21 |
| I50 | Heart failure | 48.25 |
| I95 | Hypotension | 15.42 |
| J18 | Pneumonia, unspecified organism | 10.85 |
| J44 | Other chronic obstructive pulmonary disease | 15.56 |
| J96 | Respiratory failure, not elsewhere classified | 24.08 |
| K21 | Gastro-esophageal reflux disease | 28.88 |
| N17 | Acute kidney failure | 37.84 |
| N18 | Chronic kidney disease (CKD) | 35.05 |
| N39 | Other disorders of urinary system | 12.25 |
| N40 | Benign prostatic hyperplasia | 9.80 |
| Y83 | Surgical operation and other surgical procedures as cause of abnormal reaction or later complication | 11.90 |
| Y92 | Place of occurrence of the external cause | 34.06 |
| Z66 | Do not resuscitate | 16.67 |
| Z68 | Body mass index (BMI) | 21.61 |
| Z79 | Long term (current) drug therapy | 48.84 |
| Z85 | Personal history of malignant neoplasm | 22.80 |
| Z86 | Personal history of certain other diseases | 22.22 |
| Z87 | Personal history of other diseases and conditions | 36.47 |
| Z95 | Presence of cardiac and vascular implants and grafts | 25.19 |
| Z99 | Dependence on enabling machines and devices, not elsewhere classified | 11.78 |

Table 3 provide a detailed description of the dataset used in this work.

Table 3: **Dataset Details**. Breakdown of each dataset, including study counts per split, total videos, average number of views per study, and original image dimensions. The average view count indicates the average number of dinstict available views per study.

|  | # Studies | | | Total # | Avg. # | Original |
|---|---|---|---|---|---|---|
|  | **Train** | **Val** | **Test** | **Videos** | **Views** | **Image Size** |
| **MIMIC-IV-ECHO** | 6'652 | 223 | 239 | 173'609 | 24.4 | $708 \times 1016$ |
| **EchoNet-Dynamic** | 7'465 | 1'288 | 1'277 | 10'030 | 1 | $112 \times 112$ |
| **EchoNet-Pediatrics** | 3'518 | 442 | 507 | 7'810 | 1.75 | $112 \times 112$ |

## 5.2 EXPERIMENTAL SETUP

We pretrain all models on MIMIC-IV-ECHO for 2'000 epochs. Frames are resized to $224 \times 224$ and divided into patches of size $14 \times 14$. For each study, we sample 8 views, and for each view we sample 8 equally spaced frames within a temporal window of 32 frames. We use a ViT-Base encoder and a ViT-Tiny decoder and the LA module consists of 3 layers. During training, we apply simple data augmentations including random cropping and rotation, and data normalization. We compare four models: LAMAE, Video-LAMAE, and the baselines VideoMAE and MAE. All models share identical training settings and the only difference between LAMAE and MAE, and between Video-LAMAE and VideoMAE, is the inclusion of the LA module. All hyperparameters are described in Table 4.

Table 4: Training and Model Hyperparameters

| Hyperparameter | LAMAE | Video-LAMAE | Image-MAE | VideoMAE |
|---|---|---|---|---|
| Image size | | | 224 | |
| Patch size (spatial) | | | 14 | |
| Number of views | | | 8 | |
| Number of frames | | | 8 | |
| Time patch size | N/A | 1 | N/A | 1 |
| Encoder embedding dim | | | 768 | |
| Encoder layers | | | 12 | |
| Encoder heads | | | 12 | |
| Decoder embedding dim | | | 192 | |
| Decoder layers | | | 4 | |
| Decoder heads | | | 3 | |
| Mask ratio | | | 0.875 | |
| Augmentation | | Random crop + rotation (scale [0.6, 1.0], ratio [0.9, 1.1]) | | |
| Latent attention encoder layers | 3 | 3 | N/A | N/A |
| Latent attention encoder heads | 12 | 12 | N/A | N/A |
| Batch size | | | 16 | |
| Base learning rate | | | 1e-4 | |
| Optimizer | | | AdamW | |
| AdamW $\beta_1$ | | | 0.9 | |
| AdamW $\beta_2$ | | | 0.999 | |
| Learning rate schedule | | Linear warmup + cosine decay | | |
| Warmup epochs | | | 10 | |
| Warmup start factor | | | 0.5 | |
| Number of epochs | | | 1600 | |
| Number of nodes | | | 4 | |
| GPUs per node | | | 4 | |
| GPU type | | | NVIDIA GH200 | |

## 5.3 METRICS

We evaluate performance using the Area Under the Receiver Operating Characteristic curve (AUROC) and the F1 score. The F1 score is defined as the harmonic mean of precision and recall:

$$F_1 = 2 \cdot \frac{\text{Precision} \cdot \text{Recall}}{\text{Precision} + \text{Recall}} = \frac{2TP}{2TP + FP + FN} \tag{4}$$

where $TP$, $FP$, and $FN$ represent true positives, false positives, and false negatives, respectively.

The AUROC measures the model's ability to discriminate between classes across all possible thresholds by calculating the area under the curve plotted as the True Positive Rate (TPR)

$$\text{TPR} = \frac{TP}{TP + FN} \tag{5}$$

against the False Positive Rate (FPR)

$$\text{TPR} = \frac{TP}{TP + FN} \tag{6}$$

Since our task involves multi-label classification of imbalanced ICD-10 codes, we report macro-averaged results to ensure each diagnosis contributes equally to the final score.

## 5.4 ADDITIONAL RESULTS

Given the heterogeneity of the ICD-10 codes considered, many diagnoses in the selected set are weakly related, or entirely unrelated to cardiac structure and function observable in echocardiograms. Examples include *K21: Gastro-esophageal reflux disease*, *N39: Other disorders of urinary system*, and *F32: Depressive episode*, which are unlikely to be predictable from imaging data alone. For that reason we only show the results for the top 10 codes, as decays beyond that point. Figure 3 shows the detailed F1 and AUROC scores for both full finetuning and frozen backbone setups.

Figure 3: AUROC and F1 scores for the top 10 ICD-10 codes.

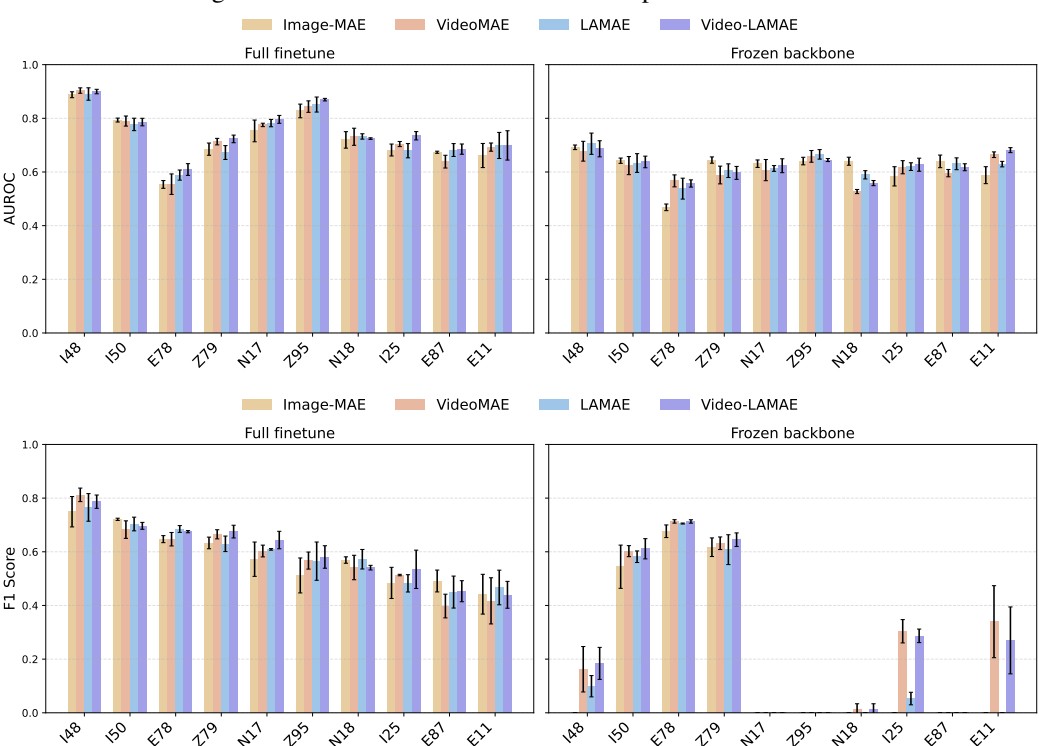

