# OpenReview forum: "Beyond Independent Frames: Latent Attention Masked Autoencoders for Multi-View Echocardiography"
_ICLR.cc/2026/Workshop/FM4Science — ICLR 2026 Workshop FM4Science Poster_

### Official Review · Reviewer_N5g1 · 2026-02-16
**Modest but meaningful gains from latent cross-view attention in echocardiography MAEs**

**Rating:** 6
**Confidence:** 2

**Review:**

## Summary

The paper proposes **Latent Attention Masked Autoencoder (LAMAE)**, a multi-view extension of masked autoencoders designed for echocardiography video representation learning. The core idea is to insert a **latent attention (LA) module** between the encoder and decoder to enable cross-view and cross-frame token interaction prior to reconstruction. The motivation to move beyond independent frame or view processing is well-justified for cardiac imaging, where clinical studies typically consist of multiple videos acquired from different views capturing complementary anatomical information.

The architectural modification is clean and well-controlled, and the comparison against **Image-MAE** and **VideoMAE** is appropriately structured. The pediatric transfer experiment is particularly notable, showing a meaningful improvement under domain shift. However, the gains are modest in most settings, several claims are overstated, and important ablations are missing.

---

## Strengths

### 1. Clean and Controlled Architectural Contribution

The LA module is inserted between the encoder $E_{\phi}$ and the decoder $D_{\theta}$ while maintaining identical hyperparameters and training protocols relative to the MAE baselines. As detailed in **Appendix Table 4**, all models share the same ViT-Base encoder and training settings, which strengthens the validity of the empirical results.

### 2. Meaningful Pediatric Transfer Result

On **EchoNet-Pediatrics**, Video-LAMAE achieves **$3.92 \pm 0.05$ MAE** compared to **$4.24 \pm 0.13$** for VideoMAE. The reported confidence intervals do not overlap, suggesting a meaningful improvement that aligns with the hypothesis that cross-view aggregation improves robustness under domain shift, especially when anatomical differences challenge models trained on adult data.

### 3. Proper Sanity Check on Single-View Dataset

On **EchoNet-Dynamic**, a single-view dataset, VideoMAE and Video-LAMAE achieve nearly identical performance (**4.34 MAE**). This controlled result supports the claim that the LA module provides benefit specifically in multi-view settings rather than introducing spurious gains.

---

## Concerns

### 1. Overstated Claims of Consistent Superiority

The manuscript claims that Video-LAMAE "consistently achieves higher AUROC and F1 scores" than standard MAE counterparts. This is not strictly true. In the **frozen-backbone setting (Table 1)**, VideoMAE achieves a slightly higher F1 score (**$0.28 \pm 0.03$**) than Video-LAMAE (**$0.27 \pm 0.03$**). While the difference is small, the claim of consistency should be softened.

### 2. Modest Improvements in Core Metrics

In the primary ICD prediction task, improvements are incremental. For example, AUROC increases from **0.73** (VideoMAE) to **0.75** (Video-LAMAE) under full fine-tuning. While positive, the magnitude of the improvement should be interpreted in the context of the relatively strong VideoMAE baseline. The manuscript would benefit from reporting statistical tests rather than relying solely on overlapping standard deviations.

### 3. Noisy Evaluation Metric (ICD-10 Codes)

The use of 40 ICD-10 codes as the primary downstream task introduces label noise. The appendix acknowledges that several codes, such as **F32 (Depressive episode)** and **K21 (Gastro-esophageal reflux)**, are unlikely to be predictable from echocardiography alone. Including these labels in macro-averaged results may dilute the signal. Reporting a secondary metric restricted to cardiovascular-related codes would strengthen the evaluation.

---

## Questions for the Authors

- **Ablation of Latent Attention vs. Global Pooling:** Could the authors provide a comparison where the LA module is replaced with simple mean pooling of the latent tokens? This would clarify whether gains stem from the attention mechanism or simply from increased view coverage.

- **Impact of View Count Scaling:** The protocol uses 8 views. How does performance scale as the number of available views decreases (e.g., $N=1,2,4,8$)? This is particularly relevant for incomplete clinical studies as noted in Table 3.

- **Necessity of Latent Space Masking:** The architecture applies additional random masking in the latent space. What is the impact of this step compared to standard patch-level masking alone?

- **Clarification on Table 1 Inconsistency:** In the frozen-backbone setting, VideoMAE outperforms Video-LAMAE in F1 score. How do the authors reconcile this with the claim of consistent superiority?

- **Cardiac-Specific Evaluation:** Have the authors considered reporting a macro-averaged score restricted only to codes directly related to cardiovascular structure?

---

## Overall Assessment

The contribution is incremental but technically sound. The pediatric transfer results provide meaningful evidence that latent cross-view attention improves robustness under domain shift. While several ablations and clarifications are needed to strengthen the empirical case, the work represents a reasonable architectural contribution for a workshop venue.

---

### Official Review · Reviewer_iaXk · 2026-02-21
**This paper proposes  practical solution to the multi-view data integration problem in echocardiography. By introducing a Latent Attention module to the Masked Autoencoder architecture, the model learns to synthesize cardiac structure and function across disparate echocardiography views. It provides a solid foundational benchmark on MIMIC-IV-ECHO and demonstrates excellent transferability to pediatric cohorts.**

**Rating:** 6
**Confidence:** 4

**Review:**

Strong Points:
1. The paper demonstrates that the results on specialized real datasets provide strong empirical validation of the latent attention module's utility in real-world domain shifts.
2. Although the core components (ViT encoders/decoders, standard MAE objective, self-attention) are well-established primitives in the ML community, the application in the specific domain shows novelty.
3. Providing the first results for ICD-10 code prediction from MIMIC-IV-ECHO videos serves as a strong, novel benchmark for the community. However, the multi-label classification of ICD-10 codes yields relatively low absolute F1 scores (e.g., ~0.27 in frozen settings). This indicates that while the representations are better than the baselines, the task remains fundamentally extremely noisy and difficult to solve using imaging data alone.


Weak Points:
1. Limited cross-view representation learning: The masking strategy relies on randomly masking subsets of image patches but does not explicitly force the model to reconstruct one view from another view's context during pretraining.
2. Lack of discussion on computational overhead: Since self-attention mechanisms scale quadratically with sequence length, concatenating tokens across 8 views and 8 frames could introduce significant memory/compute bottlenecks, which is not thoroughly analyzed.

---

### Official Review · Reviewer_FdWk · 2026-02-23
**Beyond Independent Frames: Latent Attention Masked Autoencoders for Multi-View Echocardiography**

**Rating:** 7
**Confidence:** 3

**Review:**

Summary:

This paper proposes Latent Attention Masked Autoencoder, a masked-autoencoding pretraining framework designed for multi-view echocardiography studies, where multiple videos/views per study should exchange information rather than being processed as independent frames/clips. The key idea is a latent attention module inserted between encoder and decoder to aggregate tokens across sampled views/frames in latent space while keeping a standard MAE-style reconstruction objective.

Pros:
1. The problem framing matches real echocardiography workflows (multi-view studies) and addresses a real gap in “independent frame/clip” SSL.
2. The LA module is a clean, architecture-level inductive bias that directly targets cross-view/frame information fusion during pretraining, and it is described clearly.
3. Empirical results show consistent improvements over MAE/VideoMAE for ICD-10 prediction under full finetuning and a reasonable ablation structure.
4. Transfer results are the strongest evidence for the multi-view prior: Video-LAMAE improves pediatric LVEF prediction and looks more robust under domain shift.

Cons:
1. The ICD-10 prediction task is intrinsically noisy/weakly related to echo content for many codes; even the paper notes that many selected codes are unlikely to be predictable from imaging alone, which complicates interpretation of headline metrics.
2. Comparisons are limited to MAE/VideoMAE-style baselines; it is hard to place the method vs. recent echocardiography-specific foundation models or multi-view fusion methods beyond those two.
3. Reporting focuses on top-10 ICD codes due to performance decay; the practical utility of the full 40-code setup is unclear.

---

### Meta-Review · Area_Chair_n4Ud · 2026-02-28

**Recommendation:** Accept (Poster)
**Confidence:** 4

**Metareview:**

This paper introduces LAMAE, a masked autoencoder architecture augmented with a latent attention module to aggregate information across multi-view echocardiography studies. Reviewers find the problem framing well aligned with real clinical workflows and the architectural contribution clean and well controlled. Empirical results show modest but consistent improvements over MAE/VideoMAE baselines, with particularly encouraging transfer performance from adult to pediatric cohorts.

Concerns include modest absolute gains, noisy ICD-10 evaluation labels, limited ablation of the latent attention mechanism, and lack of computational overhead analysis. Some claims of consistent superiority should be softened.

---

### Decision · Program_Chairs · 2026-03-03

Accept (Poster)